# Intelligent Interference Management in UAV-Based HetNets [†]

**Simran Singh [1,\*], Abhaykumar Kumbhar [2], İsmail Güvenç [1] and Mihail L. Sichitiu [1]**

[1] Department of Electrical and Computer Engineering, North Carolina State University, Raleigh, NC 27695, USA; iguvenc@ncsu.edu (İ.G.); mlsichit@ncsu.edu (M.L.S.)

[2] Motorola Solutions, Inc., Plantation, FL 33322, USA; akumb004@fiu.edu

[\*] Correspondence: ssingh28@ncsu.edu

[†] This paper is an extended version of our conference paper: Singh, S.; Kumbhar, A.; Güvenç, İ.; Sichitiu, M.L. Distributed Approaches for Inter-Cell Interference Coordination in UAV-Based LTE-Advanced HetNets. In Proceedings of the IEEE 88th Vehicular Technology Conference (VTC-Fall), Chicago, IL, USA, 27–30 August 2018.

**Abstract:** Unmanned aerial vehicles (UAVs) can play a key role in meeting certain demands of cellular networks. UAVs can be used not only as user equipment (UE) in cellular networks but also as mobile base stations (BSs) wherein they can either augment conventional BSs by adapting their position to serve the changing traffic and connectivity demands or temporarily replace BSs that are damaged due to natural disasters. The flexibility of UAVs allows them to provide coverage to UEs in hot-spots, at cell-edges, in coverage holes, or regions with scarce cellular infrastructure. In this work, we study how UAV locations and other cellular parameters may be optimized in such scenarios to maximize the spectral efficiency (SE) of the network. We compare the performance of machine learning (ML) techniques with conventional optimization approaches. We found that, on an average, a double deep Q learning approach can achieve 93.46% of the optimal median SE and 95.83% of the optimal mean SE. A simple greedy approach, which tunes the parameters of each BS and UAV independently, performed very well in all the cases that we tested. These computationally efficient approaches can be utilized to enhance the network performance in existing cellular networks.

**Keywords:** artificial intelligence; double deep Q learning; FeICIC; HetNets; LTE-advanced; UAV





## 1. Introduction

An unmanned aerial vehicle (UAV) heterogeneous network (HetNet) consists of conventional stationary ground macro base stations (MBSs), supplemented by mobile UAV base stations (UABSs) and cells on wheels [1]. The agility of UAVs coupled with their ability to carry radios and communicate wirelessly has led their adoption in various applications to address network congestion and in public safety communications as a temporary substitute for damaged communication infrastructure. In the aftermath of hurricane Maria in 2017, ground base stations were destroyed and AT&T used UAVs to temporarily restore wireless voice, text, data, and multimedia services [2].

UABSs can complement existing MBSs in a UAV HetNet by providing wireless coverage to user equipment (UE) that are near cell borders, coverage holes, or far away from the MBS as shown in Figure 1. In this scenario, UABS 1 serves UE 5, while UABS 2 serves UE 6. These are the UEs that are far away from MBS 1, and thus these UABSs extends the coverage area of the network. On the other hand, though UABSs transmit at a lower power than MBSs, UABSs may cause interference to MBSs and vice-versa. Thus, the transmit powers of MBSs and UABSs needs to be adjusted accordingly to ensure sufficient signal to noise ratio (SINR) for all UEs.

As UABSs are generally powered by battery and have limited computational capabilities, any network optimization algorithms that are developed need to have low computational complexity. Additionally, the solutions should also have low time complexity so that the UABS can respond quickly to changes in cellular demand. Taking into

account these constraints, we attempt to reduce the computational and time complexity of our proposed algorithms.

In this work, we develop computationally efficient algorithms to maximize the mean and median spectral efficiency (SE) in a UAV HetNet and evaluate the performance of these algorithms. This works extends a previous work written by the same authors [3]. In [3], we optimized the fifth percentile of spectral efficiency (5pSE) in a UAV-HetNet by employing a greedy algorithm and an artificial intelligence (AI) approach based on deep Q learning (DQN). In [3], it was observed that an AI approach, somewhat surprisingly, failed to find the optimum solution and was always out-performed by the greedy approach. In this work, we present an alternate AI solution, which is based on a double deep Q learning algorithm (DDQN) and uses a single AI agent to model all MBSs and UABSs in the UAV HetNet.

This single AI agent chooses the values of all relevant network parameters, unlike in [3], where each macro base station (MBS) and UAV base station (UABS) was modeled as a separate AI agent. Its performance is evaluated against the optimal exhaustive search and the computationally efficient sequential algorithm introduced in [3], which optimizes BS and UAV parameters in a greedy manner. Compared to [3], the performance of the AI approach is now closer to the optimal. Specifically, the AI algorithm can achieve 93.46% of the optimal, when maximizing the median SE and 95.83% of the optimal, when maximizing the mean SE.

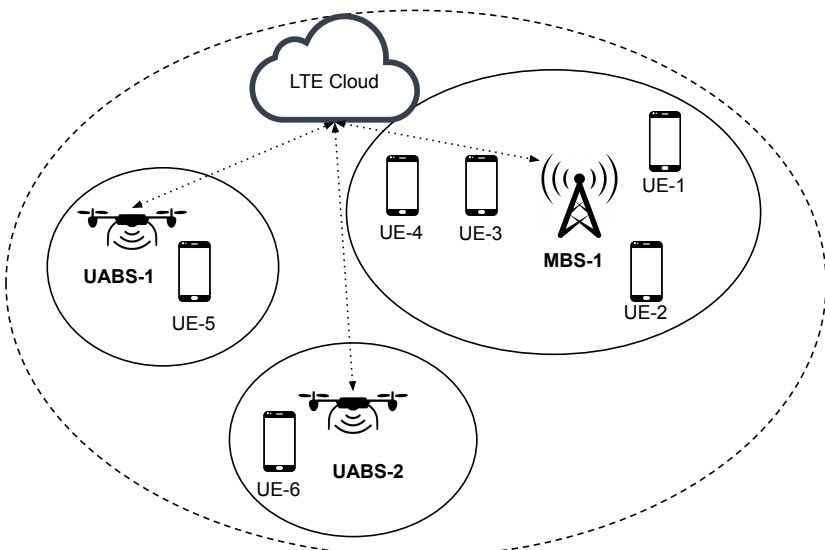

**Figure 1.** A UAV-assisted HetNet composed of UABSs and an MBS where certain UEs are offloaded to UABSs to improve the coverage and fairness.

## 2. Literature Review

Various studies in the literature have explored how UAVs can be used to supplement cellular networks. Approaches to calculate the best UABS position and trajectory have been studied in [4–10]. The authors in [4] calculated the best trajectory for a single UABS, assumed that the MBS and UABS operated at orthogonal frequencies, and thus the effect of interference was ignored. Greedy and unsupervised learning-based algorithms to position a fleet of UAVs to maximize aggregate user received power were presented in [5], while a sequential spiral algorithm was presented in [6].

However, none of these works considered interference between UAVs and BSs. Interference between UAVs and BSs was taken into account in [7,8]. Interference was minimized by using a statistical interference map and orthogonal frequency assignment in [7] to select optimal UAV locations, while multi-antenna beam-forming techniques were used in [8]. None of the studies leveraged LTE 3GPP Release-10/11 interference management techniques, nor any machine learning techniques. Machine learning techniques were used to predict potential network congestion and deploy UAVs accordingly in [10].

Interference between UAVs and BSs was assumed to be minimized using orthogonal frequency assignments and beamforming techniques rather than with LTE 3GPP Release-10/11 techniques. The authors in [11] followed the intuition that UAVs should be placed at coverage holes to maximize coverage, and used machine learning to locate coverage holes. A greedy approach to position UAVs was used in [12] to maximize the number of users covered, while ensuring that the QoS requirements of the users are satisfied.

Game-theory-based approaches were used in [13] The use of a UAV to provide wireless service to an Internet of Things (IoT) network was studied in [13], wherein IoT nodes harvest energy from a UAV before transmitting data on the uplink to the UAV. The nodes form coalitions with one node acting as the coalition head. The optimal trajectory of the UAV was calculated to maximize the energy available to the IoT coalition heads.

Novel approaches to optimally partition a geographical area into UABS and MBS cells were proposed in [14,15]. While [14] used optimal transport theory with the aim of minimizing the transmission delay for all users in the area, ref. [15] used a neural-based cost function in which user demand patterns are used to assign a cost and density to each area. Both studies, however, did not tackle interference mitigation challenges explicitly. Interference mitigation using 3GPP Release-10 enhanced inter-cell interference coordination techniques (eICIC) and Release-11 further enhanced inter-cell interference coordination (FeICIC) techniques in HetNets were studied in [16–19]. While [16] jointly optimized the ICIC parameters, the UE cell association rules, and the spectrum resource sharing between the macro and pico cells, it did not use 3GPP Release-11 FeICIC or cell range expansion (CRE) techniques.

Moreover, ref. [16] only studied LTE HetNets and not UAV HetNets. LTE UAV HetNets were also evaluated in [20], which optimized the allocation of LTE physical resource blocks in addition to the UAV position in order to maximize coverage. However, LTE interference management techniques were not utilized. In [17], the authors developed a stochastic-geometry-based framework to study and compare the effectiveness of 3GPP FeICIC techniques and eICIC techniques, but [17] also did not study UAV HetNets. The use of 3GPP Release-10/11 techniques along with UABS mobility in UAV HetNets was evaluated in [19]. However, this study did not individually optimize the 3GPP ICIC parameters, but rather applied the same ICIC parameter values to each MBS and UABS, which is sub-optimal, as we demonstrate. An overview of the existing literature that is related to our work is presented in Table 1.

To the best of our knowledge, low complexity approaches to optimize 3GPP Release-10/11 interference management parameters in UAV HetNets have not been studied in the literature. Our contribution is that we propose a greedy algorithm and a double deep Q learning-based algorithm, to individually optimize 3GPP Release-10/11 interference coordination parameters and UABS position in order to maximize the mean and median SE. We also compare these two computationally efficient algorithms with an optimal but computationally complex brute force algorithm.

**Table 1.** Comparison with related works.

| Ref. | Applicable to HetNets | UAVs Used as BSs | 3GPP Interference Management | Optimization Variables | Optimization Goal | Optimization Algorithm(s) |
|---|---|---|---|---|---|---|
| [4] | ✓ | ✓ | ✗ | Bandwidth allocated to the UAV, user-cell association, radius of UAV's trajectory | Maximize the minimum throughput across all users | Bisection search |
| [5] | ✓ | ✓ | ✗ | Number and position of UAVs | Minimize the number of UAVs, and maximize aggregate user received power | Greedy approach, a novel unsupervised learning approach |
| [6] | ✗ | ✓ | ✗ | Number and position of UAVs | Minimize the number of UAVs | A sequential algorithm that places UAVs along a spiral |

**Table 1.** *Cont.*

| Ref. | Applicable to HetNets | UAVs Used as BSs | 3GPP Interference Management | Optimization Variables | Optimization Goal | Optimization Algorithm(s) |
|---|---|---|---|---|---|---|
| [8] | ✓ | ✓ | ✗ | UAV position | Minimize the distance travelled by each UAV while maximizing the LoS MIMO channel capacity | Exhaustive search, gradient descent |
| [9] | ✓ | ✓ | ✗ | User-cell associations, downlink power allocations and UAV position | Maximize network sum rate, subject to a constraint on received SINR | Hybrid fixed-point iteration, particle swarm optimization |
| [10] | ✓ | ✓ | ✗ | Location and area served by each UAV | Minimize total downlink power | Machine learning framework based on a Gaussian mixture model and weighted expectation maximization |
| [14] | ✓ | ✓ | ✗ | Area served by the BS and the UAV | Minimize mean user transmission delay | Optimal transport theory |
| [16] | ✓ | ✗ | ✓ | Sub-frame radio resource sharing among macro BSs and pico cells, user-cell association | Maximize weighted proportional proportional fair throughput | Dual-based approach to solve a relaxed NLP followed by integer rounding |
| [17] | ✓ | ✗ | ✓ | 3GPP FeICIC parameters | Maximize aggregate capacity and proportional fairness among users | Exhaustive search |
| [19] | ✓ | ✓ | ✓ | 3GPP FeICIC parameters. Parameters *not* optimized individually for each BS and UAV. | Maximize fifth percentile of spectral efficiency (SE) | Exhaustive search, genetic algorithm, and elitist harmony search |
| [13] | ✓ | ✓ | ✗ | UAV trajectory and uplink transmission of each Internet of Things (IoT) node | Maximize energy availability of IoT coalition heads | Exhaustive search for UAV trajectory and non-cooperative game theory-based approach to calculate uplink transmission power |
| [21] | ✓ | ✓ | ✗ | Trajectory of multiple UAVs, association of devices to UAVs, and uplink power | Minimize total uplink transmit power | Main optimization problem decomposed into two sub-problems, which are solved iteratively together |
| [20] | ✓ | ✓ | ✗ | UAV position and allocation of LTE physical resource blocks | Maximize number of users covered, while satisfying their delay requirements | A heuristic-based approach, which achieves near-optimal performance |
| [12] | ✓ | ✓ | ✗ | UAV position | Maximize number of users covered, while satisfying their QoS requirements | Greedy approach |
| [11] | ✓ | ✓ | ✗ | UAV position | Maximize number of users covered | Reinforcement learning used to discover network coverage holes, where UAVs are then positioned |
| This work | ✓ | ✓ | ✓ | 3GPP FeICIC parameters. Parameters optimized individually for each BS and UAV. | Mean and median SE | Double deep Q learning, greedy approach, exhaustive search |

## 3. System Model

We consider a HetNet with MBSs and UABSs operation in two tiers, within a simulation area of $l \times l$ square meters as shown in Figure 2. MBSs and UEs are randomly

distributed in this area, according to a Poisson point process with the intensities $\lambda_{\text{mbs}}$ and $\lambda_{\text{ue}}$, respectively. The number of MBSs and UEs in the simulation can be calculated as $N_{\text{mbs}} = \lambda_{\text{mbs}} \times l^2$ and $N_{\text{ue}} = \lambda_{\text{ue}} \times l^2$, respectively. The 3D locations of all the MBSs, UABSs, and UEs are represented by the matrices $\mathbf{X}_{\text{mbs}} \in \mathbb{R}^{N_{\text{mbs}} \times 3}$, $\mathbf{X}_{\text{uabs}} \in \mathbb{R}^{N_{\text{uabs}} \times 3}$, and $\mathbf{X}_{\text{ue}} \in \mathbb{R}^{N_{\text{ue}} \times 3}$, respectively. The transmission power of each MBSs is $P_{\text{mbs}}$, while that of UABSs is $P_{\text{uabs}}$. Given the antenna gains of MBS and UABS as $K$ and $K'$, respectively, the effective MBS and UABS transmission power is calculated as $P'_{\text{mbs}} = KP_{\text{mbs}}$ and $P'_{\text{uabs}} = K'P_{\text{uabs}}$, respectively.

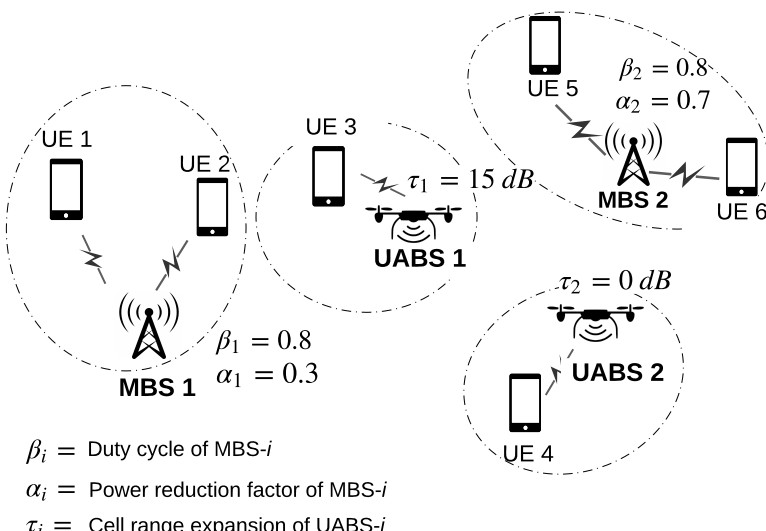

**Figure 2.** Illustration of a UAV HetNet where the ICIC parameters of all MBSs and UABSs, and the locations of UABSs are optimized individually. This approach is in contrast to some existing work, such as [22], which, while optimizing locations for each UABS individually, assigns the same ICIC parameters to all UABSs and MBSs, Reprinted with permission from Ref. [3]. Copyright 2018 IEEE.

We assume that the UABSs and MBSs exchange information over the X2 interfaces. We also consider that the downlink bandwidth available to an MBS or a UABS is shared equally among their served UEs. We assume that downlink data is always available for any UE—i.e., the downlink UE traffic buffers are always full. For an arbitrary UE $n$, where $n \in \{1, 2, \ldots, N_{\text{ue}}\}$, we define the macro-cell of interest (MOI) as the nearest MBS, and the UAV-cell of interest (UOI) as the nearest UABS.

For example, in the specific scenario shown in Figure 2, MBS 1 is the MOI for UEs 1 and 2, MBS 2 is the MOI for UEs 5 and 6, UABS 1 is the UOI of UE 2, and UABS 2 is the UOI of UE 3. We denote the reference symbol received power (RSRP) of the $n$th UE from the MOI and the UOI as $S_{\text{mbs}}(d_{mn})$ and $S_{\text{uabs}}(d_{un})$, respectively, where $d_{mn}$ is the distance from the nearest MOI, and $d_{un}$ is the distance from the nearest UOI for the $n$th UE. We use the Okumura Hata suburban propagation model without any Rayleigh or Rician fading.

An arbitrary UE $n$ is always assumed to connect to the nearest MBS or UABS, where $n \in 1, 2, \ldots, N_{\text{ue}}$. Then, for the $n$th UE, the reference symbol received power (RSRP) from the macro-cell of interest (MOI) and the UAV-cell of interest (UOI) are given by

$$S_{\text{mbs}}(d_{mn}) = \frac{P'_{\text{mbs}}}{10^{\varphi/10}}, \ S_{\text{uabs}}(d_{un}) = \frac{P'_{\text{uabs}}}{10^{\varphi'/10}}, \tag{1}$$

where $\varphi$ is the path-loss observed from MBS in dB, $\varphi'$ is the path-loss observed from UABS in dB, $d_{mn}$ is the distance from the nearest MOI, and $d_{un}$ is the distance from the nearest UOI.

The Okumura Hata path loss is a function of the carrier frequency, distance between the UE, serving cell, base station height, and UE antenna height. The path-loss (in dB) observed by the $n$th UE from MOI and UOI is given by:

$$\varphi = A + B\log(d_{mn}) + C, \tag{2}$$
$$\varphi' = A + B\log(d_{un}) + C, \tag{3}$$

where the distances $d_{mn}$ and $d_{un}$ are in km, and the factors $A$, $B$, and $C$ depend on the carrier frequency and antenna height. In a suburban environment, the factors $A$, $B$, and $C$ are given by

$$A = 69.55 + 26.16\log(f_c) - 13.82\log(h_{bs}) - a(h_{ue}), \tag{4}$$
$$B = 44.9 - 6.55\log(h_{bs}), \tag{5}$$
$$C = -2\log(f_c/28)^2 - 5.4, \tag{6}$$

where $f_c$ is the carrier frequency in MHz, $h_{bs}$ is the height of the base station in meters, and $a(h_{ue})$ is the correction factor for the UE antenna height $h_{ue}$ in meters, which is defined as

$$a(h_{ue}) = 1.1\log(f_c) - 0.7h_{ue} - 1.56\log(f_c) - 0.8. \tag{7}$$

### 3.1. SE with 3GPP Release-10/11 ICIC Techniques

In a HetNet, the MBSs transmit at higher powers and have higher ranges compared to the lower power UABSs. Nevertheless, the UABSs can extend their coverage and associate a larger number of UEs by using the cell range expansion (CRE) technique defined in 3GPP Release-8. The CRE of a UABS is defined as the factor by which UEs are biased to associate with that UABS. For example, in Figure 2, UABS 1 uses a CRE of 15 dB to force UE 4 to associate with itself. The use of CRE, however, results in increased interference to those UEs in the extended cell regions.

This interference from MBSs to UEs near the edge of range-extended UABS cells can be mitigated using time-domain-based ICIC techniques defined in 3GPP Release-10/11. These techniques require the MBS to transmit with reduced power during specific subframes on the physical downlink shared channel (PDSCH). Radio subframes with reduced power are termed coordinated subframes (CSF), and those with full power are termed uncoordinated subframes (USF).

We denote this power reduction factor by $\alpha$ where $0 \leq \alpha \leq 1$. We note that $\alpha = 1$ implies no ICIC, while eICIC techniques use $\alpha = 0$, and FeICIC techniques allow $\alpha$ to vary between 0 and 1. We use $\beta$ to denote the USF duty cycle, and hence, the CSF duty cycle is given by $(1 - \beta)$. Figure 3 shows, for the scenario depicted in Figure 2, how MBS 1 and MBS 2 use power reduction factors, $\alpha_1$ and $\alpha_2$ respectively, to reduce interference to UE 3. We note that $\alpha_1 < \alpha_2$, as MBS 2, being farther away from UE 3, can transmit at a higher power without degrading the performance of UE 3.

Individual MBSs or UABSs can schedule their UEs in USF or CSF based on the scheduling thresholds $\rho$ and $\rho'$, respectively. Then, a UE may be served either by an MOI or UOI and by the CSF or USF resources of the MOI/UOI, resulting in four different association categories. Let $\Gamma$ denote the signal to interference ratio (SIR) at the MOI-USF, $\Gamma'$ denotes the SIR at the UOI-USF, and $\tau$ denotes the CRE that positively biases the UABS SIR to expand its coverage. Then, the four different resource categories where a UE may be scheduled can be summarized as follows. If $\Gamma > \tau\Gamma'$, we associate the UE with the MOI; otherwise, we schedule it with the UOI.

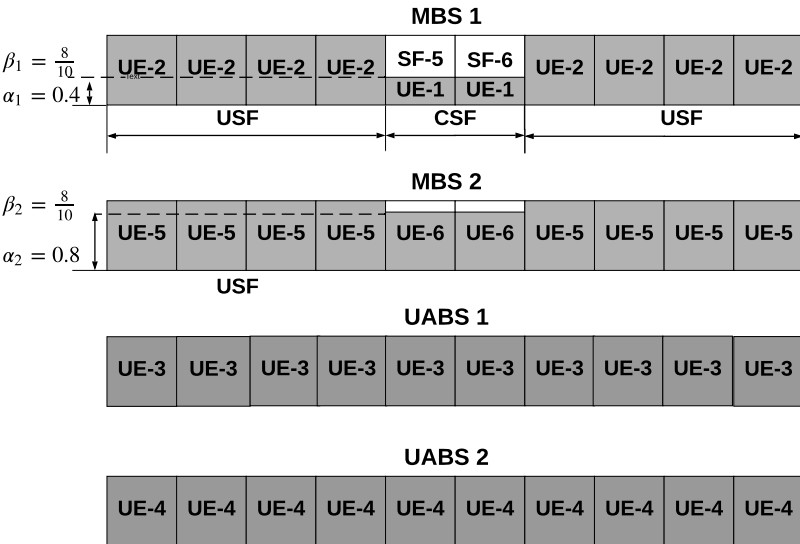

**Figure 3.** 3GPP Release-11 FeICIC with reduced power, almost blank subframes, and different power reduction factors for each MBS for the scenario in Figure 2.

The intuition behind this condition is straightforward: associate the UE with the nearest base station that gives the best SIR, taking into account the CRE. On the other hand, if $\Gamma > \rho$ or $\Gamma' \leq \rho'$, we schedule the UE in CSF, and otherwise in USF. This condition is based on the following intuition: scheduling a UE in the CSF of an MOI degrades that UE's SIR, whereas scheduling a UE in the CSF of a UOI improves that UE's SIR. Thus, the "stronger" UEs that have sufficiently high SIRs and are close to the MBS should be scheduled in the CSFs of that MBS, as these "stronger" UEs can take the performance hit. Similarly, the "weaker" UEs that have low SIRs and are close to the cell edge of a UABS should be scheduled in the CSF of that UABS, as they need to be protected as a priority.

Using this framework of eICIC and FeICIC and following an approach similar to that of [17,22] for a HetNet, the SIR ($\Gamma$, $\Gamma_{\mathrm{csf}}$, $\Gamma'$, $\Gamma'_{\mathrm{csf}}$) and the SE ($C_{\mathrm{usf}}^{\mathrm{mbs}}$, $C_{\mathrm{csf}}^{\mathrm{mbs}}$, $C_{\mathrm{usf}}^{\mathrm{uabs}}$, $C_{\mathrm{csf}}^{\mathrm{uabs}}$) experienced by an arbitrary UE $n$ can be defined for four different scenarios as follows:

(1) UE associated with MOI and scheduled in USF:

$$\Gamma = \frac{S_{\mathrm{mbs}}(d_{mn})}{S_{\mathrm{uabs}}(d_{un}) + Z} \qquad C_{\mathrm{usf}}^{\mathrm{mbs}} = \frac{\beta \log_2(1 + \Gamma)}{N_{\mathrm{usf}}^{\mathrm{mbs}}}. \qquad (8)$$

(2) UE associated with MOI and scheduled in CSF:

$$\Gamma_{\mathrm{csf}} = \frac{\alpha S_{\mathrm{mbs}}(d_{mn})}{S_{\mathrm{uabs}}(d_{un}) + Z} \qquad C_{\mathrm{csf}}^{\mathrm{mbs}} = \frac{(1 - \beta) \log_2(1 + \Gamma_{\mathrm{csf}})}{N_{\mathrm{csf}}^{\mathrm{mbs}}}. \qquad (9)$$

(3) UE associated with UOI and scheduled in USF:

$$\Gamma' = \frac{S_{\mathrm{uabs}}(d_{un})}{S_{\mathrm{mbs}}(d_{mn}) + Z'} \qquad C_{\mathrm{usf}}^{\mathrm{uabs}} = \frac{\beta \log_2(1 + \Gamma')}{N_{\mathrm{usf}}^{\mathrm{uabs}}}. \qquad (10)$$

(4) UE camped on UOI and scheduled in CSF:

$$\Gamma'_{\mathrm{csf}} = \frac{S_{\mathrm{uabs}}(d_{un})}{\alpha S_{\mathrm{mbs}}(d_{mn}) + Z'} \qquad C_{\mathrm{csf}}^{\mathrm{uabs}} = \frac{(1 - \beta) \log_2(1 + \Gamma'_{\mathrm{csf}})}{N_{\mathrm{csf}}^{\mathrm{uabs}}}. \qquad (11)$$

In (8)–(11), $Z$ and $Z'$ are, respectively, the interference at a UE from all the MBSs and the UABSs except the UOI and except the MOI. They have the same meaning but different values.

### 3.2. Performance Metrics

The objective of our study is to design algorithms to calculate the best FeICIC and eICIC parameters individually for all MBSs and UABSs as well as the positions of the UABSs to, thus, maximize our performance metric. We evaluate two performance metrics: the mean SE and median SE. The algorithm should find the best state, $\mathbf{S}'$, out of all possible states $\mathbf{S}$ such that:

$$\mathbf{S}' = \arg_{\mathbf{S}} \max_{SE} \mathbf{C} (\mathbf{S}), \tag{12}$$

where $C(.)$ denotes the function that calculates the mean or median SE over the whole network area for a given state $\mathbf{S} = [\mathbf{X}_{\text{uabs}}, \mathbf{S}_{\text{mbs}}^{\text{ICIC}}, \mathbf{S}_{\text{uabs}}^{\text{ICIC}}]$. First, we calculate the SE for each UE as per Equations (1)–(4). As defined previously, $\mathbf{X}_{\text{uabs}}$ is the matrix representing the location of the $N_{\text{uabs}}$ UABSs in three dimensions, $\mathbf{S}_{\text{mbs}}^{\text{ICIC}} = [\boldsymbol{\alpha}, \boldsymbol{\beta}, \boldsymbol{\rho}] \in \mathbb{R}^{N_{\text{mbs}} \times 2}$ is a matrix that captures individual ICIC parameters for each MBS, and $\mathbf{S}_{\text{uabs}}^{\text{ICIC}} = [\boldsymbol{\tau}, \boldsymbol{\rho}'] \in \mathbb{R}^{N_{\text{uabs}} \times 2}$ is a matrix that captures the individual ICIC parameters for each UABS. The vectors $\boldsymbol{\alpha} = [\alpha_1, \ldots, \alpha_{N_{\text{mbs}}}]^T$ and $\boldsymbol{\rho} = [\rho_1, \ldots, \rho_{N_{\text{mbs}}}]^T$ capture the power reduction factors and scheduling thresholds, respectively, of each MBS. On the other hand, for each UABS, $\boldsymbol{\tau} = [\tau_1, \ldots, \tau_{N_{\text{uabs}}}]^T$ and $\boldsymbol{\rho}' = [\rho_1', \ldots, \rho_{N_{\text{uabs}}}']^T$ denote the CRE and scheduling threshold, respectively.

### 3.3. Parameter Optimization

The SE is a function of $\boldsymbol{\alpha}$, $\boldsymbol{\beta}$, $\boldsymbol{\tau}$, $\boldsymbol{\rho}$, $\boldsymbol{\rho}'$, and $\mathbf{X}_{\text{uabs}}$, which are our optimization space. Either these parameters can be optimized for individual MBSs and UABSs, or the same value of each parameter can be used for all MBS and UABS, which is sub-optimal but computationally less complex.

To show that optimizing the above parameters individually for each MBS and UABS gives a better performance than optimizing the ICIC parameters jointly, we consider the hypothetical situation depicted in Figure 2. Here, UE 3 is the critical UE to be protected from interference. Intuitively, as MBS 1 is closer to UE 3 compared to MBS 2, it is desirable for MBS 1 to transmit at a lower power during CSFs. Mathematically, $\alpha_1 = 0.4 < \alpha_2 = 0.8$. As UE 4, which is served by UABS 2, is farther away from all the MBSs, UABS 2 does not have to use a large CRE to encourage UE 4 to associate with itself. In contrast, UABS 1 would have to use a larger CRE to encourage UE 3 to associate with itself. Mathematically, $\tau_1 = 15\,\text{dB} > \tau_2 = 0\,\text{dB}$. Therefore, optimizing the parameters individually for each MBS and UABS gives better performance.

The large size of the search space can be appreciated by referring to Tables 2 and 3, which list the range and size of the parameters to be optimized. In this table, the parameters $\Delta_\alpha$, $\Delta_\beta$, , $\Delta_\rho$, $\Delta_{\rho'}$, $\Delta_x$, and $\Delta_y$ denote the step sizes for $\alpha$, $\beta$, $\rho$, $\rho'$, $x$ coordinate of a UABS's location, and $y$ coordinate of a UABS's location, respectively, while $\rho_{\text{low}}$ and $\rho'_{\text{low}}$ denote the lower bounds for $\rho$ and $\rho'$, respectively. Similarly, $\rho_{\text{high}}$ and $\rho'_{\text{high}}$ denote the upper bounds for $\rho$ and $\rho'$, respectively. The actual size of the parameter space is depicted in Figure 4, which shows the number of possible states in $\mathbf{S}$, over which the individual or joint optimization algorithms have to search, in order to find the best state, $\mathbf{S}'$.

Figure 4 shows that, as the geographical area and correspondingly the number of MBSs and UABSs increase, the size of the search space increases much more rapidly for individual optimization than for joint optimization. This behavior can be understood by calculating the number of possible permutations of $\alpha$ for two MBSs considered by joint and individual optimization approaches, assuming that $\alpha$ can take four different values from 0 to 1.

While the joint optimization approach will only compare the SE metric at these four values of $\alpha$, the individual optimization approach will need to compare the SE metric for $4^2 = 16$ different permutations of $\alpha$ values of the two MBSs. In this way, the parameter search size for joint optimization is agnostic to the number of MBSs or UABSs in the simulation area. The parameter search space of individual optimization, on the other hand, increases exponentially with number of MBSs and UABSs. We also observe that, for the individual optimization approach, the size of the state space exceeds a googol ($10^{100}$) of states as the simulation area increases beyond 15 km$^2$.

**Table 2.** UABS parameters and search space.

| Parameter | Range | Search Space Size |
|---|---|---|
| UABS scheduling threshold ($\rho'$) | $\rho'_{low}, \rho'_{low} + \Delta'_\rho, \rho'_{low} + 2\Delta_{\rho'} \ldots \rho'_{high}$ | $\frac{(\rho'_{high} - \rho'_{low})}{(\Delta'_\rho)}$ |
| Cell range expansion ($\tau$) | $0, \Delta_\tau, 2\Delta_\tau, \ldots \tau_{high}$ | $\frac{\tau_{high}}{\Delta_\tau}$ |
| X coordinate of UABS | $-l/2, -l/2 + \Delta_x, -l/2 + 2\Delta_x, \ldots l/2$ | $\frac{l}{\Delta_x}$ |
| Y coordinate of UABS | $-l/2, -l/2 + \Delta_y, -l/2 + 2\Delta_y, \ldots l/2$ | $\frac{l}{\Delta_y}$ |

**Table 3.** MBS parameters and search space.

| Parameter | Range | Search Space Size |
|---|---|---|
| MBS power reduction factor ($\alpha$) | $0, \Delta_\alpha, 2\Delta_\alpha, \ldots 1$ | $1/\Delta_\alpha + 1$ |
| MBS duty cycle ($\beta$) | $0, \Delta_\beta, 2\Delta_\beta, \ldots 1$ | $1/\Delta_\beta + 1$ |
| MBS scheduling threshold $\rho$ | $\rho_{low}, \rho_{low} + \Delta_\rho, \rho_{low} + 2\Delta_\rho \ldots \rho_{high}$ | $\frac{(\rho_{high} - \rho_{low})}{(\Delta_\rho)}$ |

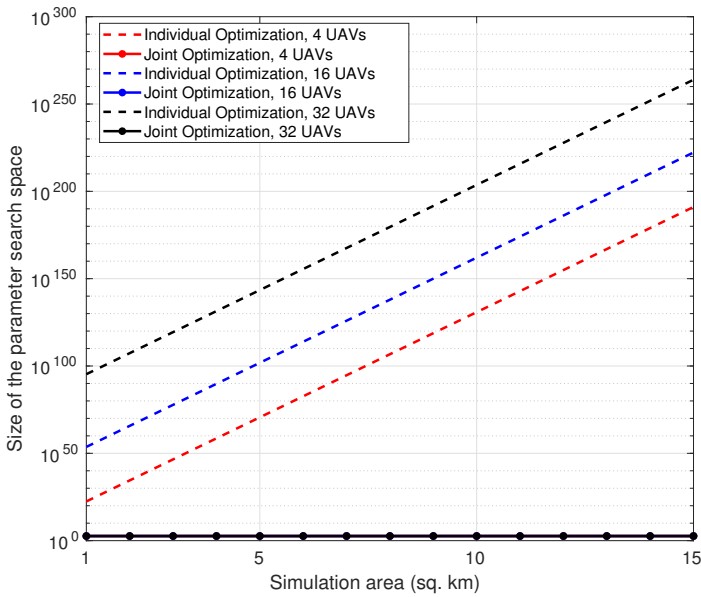

**Figure 4.** Increase in the parameter search space size as a function of the total simulation area.

## 4. UAV Location and Interference Management

We considered and compared the performance of three algorithms for FeICICs—a brute force algorithm, a greedy algorithm, and a DDQN learning algorithm. In this section,

we present the brute force and greedy algorithms, while Section IV presents the DDQN learning algorithm.

### 4.1. Brute Force Algorithm

The brute force algorithm investigates the entire search state space of all possible values of UABS locations and individual eICIC and FeICIC parameters and returns the state with the best mean or median SE. This is illustrated in Algorithm 1. Since it individually searches for all possible parameter values, it is computationally infeasible for large areas and large number of MBSs and UABSs.

---

**Algorithm 1** Brute force algorithm.

---

1: Best state, $\mathbf{S}' \leftarrow \mathbf{NULL}$
2: *bestSE* $\leftarrow -1$
3: **for all** State $\mathbf{S}$ **do**
4:    *currentSE* $\leftarrow C(\mathbf{S})$
5:    **if** *currentSE* > *bestSE* **then**
6:       *bestSE* $\leftarrow$ *currentSE*
7:       $\mathbf{S}' \leftarrow \mathbf{S}$
8:    **end if**
9: **end for**

---

### 4.2. Greedy Algorithm

In order to reduce the time complexity, we utilized a heuristic algorithm that initially assumes that there is only one UABS in the system and finds the best location, eICIC, and FeICIC parameters for this UABS. It then considers that there are two UABSs in the system, with the parameters of the first UABS set to those found earlier, and finds the best parameters for this second UABS. Similarly, it then finds the best parameters for the third UABS considering the first two UABSs to be at their earlier determined states. The algorithm continues this procedure for the given number of UABSs.

A similar approach is used to optimize the parameters of MBSs. This algorithm is summarized in Algorithm 2. As we will see in the simulation results, this algorithm, somewhat surprisingly, performs close to the brute force search, and outperforms the DDQN algorithm, to be discussed in the next section. A main reason for this is that the large number of parameters in Table 1 allows to *compensate* for a non ideal selection of parameters in earlier stages, e.g., by tuning the scheduling thresholds and CRE, non-ideal location of a UABS can still result in good SE.

---

**Algorithm 2** Greedy algorithm.

---

1: **for all** *UABS* **do**
2:    Assume all previous *UABS* to be positioned at their best location and operating at their best ICIC parameters.
3:    Find best location and FeICIC parameters for current *UABS*.
4: **end for**
5: **for all** *MBS* **do**
6:    Assume all previous *MBS* to be operating at their best ICIC parameters.
7:    Find best ICIC parameters for current *MBS*.
8: **end for**

---

## 5. Machine Learning Approach for UAV ICIC and Placement

### 5.1. Q Learning

The Q learning algorithm is one of the commonly used reinforcement learning (RL) algorithms [23]. In reinforcement learning, an agent interacts with an environment by taking actions in different states and observing the costs or rewards of the actions. The agent starts out with random actions and. eventually, by observing the rewards and by

exploring different states, it learns the best action to take in each state in order to optimize the cumulative reward. Figure 5 shows the interaction of the MBS and the UABS agent with the UAV HetNet environment considered in this paper.

Since RL and deep-learning techniques are relevant and are known to be effective in solving problems with large search spaces, it is instructive to investigate their performance in solving our UAV HetNet optimization problem. Q learning allows the agents to act and learn as follows: being in a state $s$ after selecting action $a$, and receiving the immediate cost $c$, a UABS or MBS agent $n$ updates its knowledge $Q_n(s, a)$ for the particular state-action pair through the following operation:

$$Q_n(s, a) \leftarrow (1 - \eta)Q_n(s, a) + \eta\Big[c_n + \lambda \min_a Q_n(s', a)\Big], \tag{13}$$

where $\eta$, the learning rate, is the agent's willingness to learn from the UAV HetNet environment, $\lambda$ is the discount factor, and $s'$ is the next state. Lower values of $\lambda$ give more importance to immediate rewards. We note that, conventionally, $\alpha$, rather than $\eta$, is used to denote the agent's learning rate. We use $\eta$, as, in our work, we have already used $\alpha$ to denote the ICIC power reduction factor. After the Q learning agent has been trained, and when it is being evaluated, the exploration–exploitation factor, $\epsilon$, determines the probability of the agent choosing a random action at a given state, (i.e., the agent explores), rather than choosing the optimal action, as learned by it (i.e., the agent exploits).

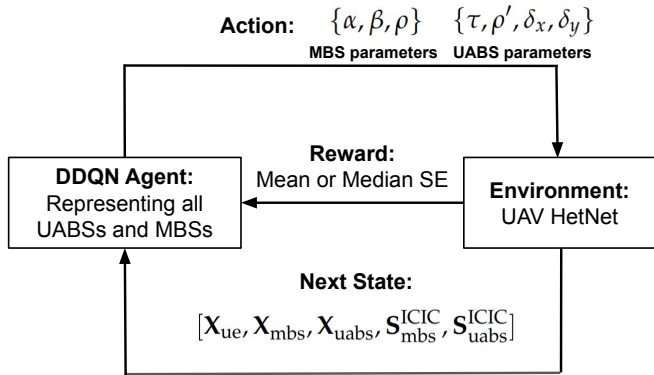

**Figure 5.** Interaction of the MBS or the UABS agent with the UAV HetNet environment for Q-learning.

The state space is a sufficient representation of the environment at a point in time, containing all the information required by the agent to choose its next action. In our case, the state space is the set:

$$[\mathbf{X}_{\text{ue}}, \mathbf{X}_{\text{mbs}}, \mathbf{X}_{\text{uabs}}, \mathbf{S}_{\text{mbs}}^{\text{ICIC}}, \mathbf{S}_{\text{uabs}}^{\text{ICIC}}], \tag{14}$$

which captures the value of each ICIC parameter for each UABS and MBS, and also the location of each MBS, UABS and UE. The action space is the union of the possible actions for UABSs and MBSs in the system. At each state, along with its position, a UABS may choose any value for its ICIC parameters, $\tau$ and $\rho'$. An MBS, similarly, may choose any value for its ICIC parameters, $\alpha$, $\beta$ and $\rho$.

Specifically, an MBS's action space is the following set:

$$\begin{aligned}
\{\alpha, \beta, \rho\} \mid \alpha &\in \{0, \Delta_\alpha, 2\Delta_\alpha, \dots 1\}, \\
\beta &\in \{0, \Delta_\beta, 2\Delta_\beta, \dots 1\}, \\
\rho &\in \{\rho_{\text{low}}, \rho_{\text{low}} + \Delta_\rho, \dots, \rho_{\text{high}}\},
\end{aligned} \tag{15}$$

and a UABS's action space is the set:

$$\{\tau, \rho', \delta_x, \delta_y\} \mid \tau \in \{0, \Delta_\tau, \ldots \tau_{\text{high}} + \Delta_\tau, \tau_{\text{high}}\},$$
$$\rho' \in \{\rho'_{\text{low}}, \rho'_{\text{low}} + \Delta_{\rho'}, \ldots \rho'_{\text{high}}\},$$
$$x_{\text{uabs}} \in \{-l/2, -l/2 + \Delta_x, -l/2 + 2\Delta_x, \ldots l/2\},$$
$$y_{\text{uabs}} \in \{-l/2, -l/2 + \Delta_x, -l/2 + 2\Delta_x, \ldots l/2\}, \quad (16)$$

where $x_{\text{uabs}}$ and $y_{\text{uabs}}$ represent the X and Y coordinate, respectively, of the UABS, as a result of the action. Our chosen reward function is the median or mean SE.

As conventional Q learning algorithms maintain a Q table, with states as rows and actions as columns and with each cell representing the Q value of a specific action in a specific state, they cannot handle infinite state spaces, as the size of the Q table would become infinite. The Q table would take an extremely long duration to converge. Conventional Q learning suffers from issues of memory complexity, computational complexity, and sample complexity [24]. As illustrated in Figure 4, the parameter space for our scenario can become extremely large, and therefore a conventional Q learning approach is not feasible. Deep learning, relying on the powerful functional approximation and representational learning properties of deep neural networks, provides us with the tools for overcoming these problems [25].

### 5.2. Deep Q Learning

DQN, hailed as the first step towards general AI—an AI that can survive in a variety of environments, came to the forefront of machine learning when it was used by DeepMind and Google to train an agent to achieve professional level scores on 49 different Atari 2600 games [26]. DQN extends Q learning by using a neural network to model the Q function, instead of using the simple Q table. One approach is to design the network to accept the state and action as the inputs and provide the corresponding Q value as the output. Another approach is for the network to accept the current state as the input and provide the Q value of each possible action as the output.

The latter approach, used in [26], is found to be better as only a single forward pass through the network is needed when we want to do a Q value update or pick the action with the highest Q value [27]. This is the architecture that we use as well. This neural network can handle infinite state spaces and also recognize common patterns between similar states. Another advantage of deep Q learning is experience replay—the neural network is retrained after each action step, enabling the agent to adapt to changes in the environment. Thus, the deep Q learning agent can learn from its experience continuously.

### 5.3. Double Deep Q Learning (DDQN)

In our simulations, we use the double DQN (DDQN) [28] architecture, which leads to better policy evaluation in the presence of many similar-valued actions and reduces over-estimation in action values compared to DQN. The DDQN approach decouples the selection of an action from the evaluation of an action. The *Q*-learning equation in (13) is then modified as:

$$Q_n(s, a) \leftarrow (1 - \eta)Q_n(s, a) + \eta\left[c_n + \lambda Q'_n(s', \min_a Q_n(s', a))\right], \quad (17)$$

where $Q'_n$ is the evaluation network that is used to evaluate the policy, while $Q'_n$ is the online network that is used to select the optimal actions. All MBSs and UABSs in the UAV HetNet are modeled by a *single* AI agent, which chooses the values of all relevant network parameters. The neural network parameters are listed in Table 4, while our DDQN approach is presented in Algorithm 3.

**Table 4.** Parameters relevant to the DDQN RL algorithm.

| Layer | Number of Neurons |
|---|---|
| Input layer | Size of state space |
| Middle layer | 512, densely connected |
| Output Layer | Size of action space |

**Algorithm 3** DDQN learning for UAV HetNets.

```
 1: while (Realization ≤ numRealizations) do
 2:     Intialize a new realization of the environment
 3:     while (NumberOfSteps ≤ numSteps) do
 4:         The agent acts, observes the reward and the new state.
 5:         Update the weights of the model neural network
 6:         Update the weights of the target neural network
 7:     end while
 8:     Choose the best state encountered so far
 9:     Preserve the neural network weights
10: end while
```

## 6. Simulation Results

The ICIC and UAV placement algorithms were evaluated in a UAV HetNet, and the parameters are as defined in Table 5. The DDQN-based AI algorithm was implemented using Intel RL Coach [29]. Intel RL Coach is a python framework, and it implements many state-of-the-art algorithms. These algorithms can be used through a set of application programming interfaces (APIs). The developer defines the environment and the optimization problem and calls the Intel RL Coach APIs to solve the problem. The UAV HetNet environment was created using python scripts, while Matlab scripts were used to simulate the path-loss model, associate UEs to BSs, and calculate the SE for a particular realization.

**Table 5.** Simulation parameters.

| Parameter | Value |
|---|---|
| MBS and UE density | 8 or 4 per km$^2$ and 100 per km$^2$ |
| MBS and UABS transmit powers | 46 dBm and 30 dBm |
| Path-loss exponent | 4 |
| Altitude of UABSs | 121.92 m (400 feet) |
| Simulation area | $0.5 \times 0.5$ km$^2$ |
| Range expansion bias in dB | 0 to 15 |
| Power reduction factor for MBS during ($\alpha$) | 0 to 1 |
| Duty cycle for the transmission of USF ($\beta$) | 0 to 1 |
| Scheduling threshold for UEs served by MBSs ($\rho$) | 35 dB or 45 dB |
| Scheduling threshold for UEs served by UABSs ($\rho\prime$) | −20 dB to −5 dB |
| Downlink frequency | 763 MHz |

The agent was trained over 300 realizations and then evaluated over 250 realizations, where the location of UEs and BSs in each realization were generated using a random Poisson point process (PPP) with UE and MBS densities as indicated in Table 5. The CDF of the median and mean SE achieved by the algorithms, calculated over multiple realizations, are shown in Figure 6a,b, respectively, when the density of MBSs is 4 MBSs per km$^2$, resulting in one MBS in the simulation area. Blue curves denote the CDF obtained using DDQN, red when using the greedy algorithm, and black when using the optimal exhaustive search.

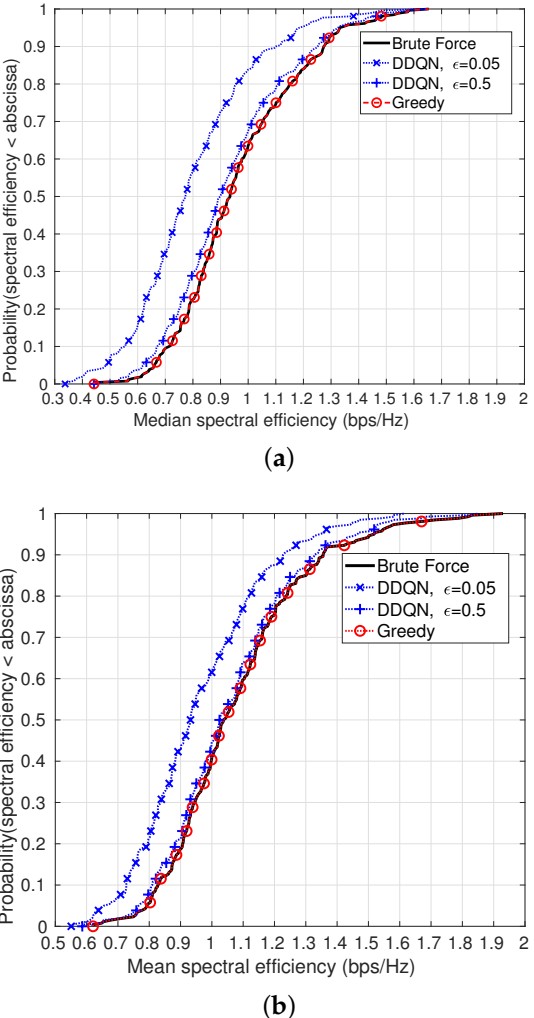

**Figure 6.** SE results for an MBS density of 4 MBS per km$^2$, which corresponds to one MBS in the simulation area. (**a**) CDF of median SE. DDQN achieves 96.56% of the optimal when $\epsilon$ = 0.5 and 82.85% of the optimal when $\epsilon$ = 0.05. (**b**) CDF of mean SE. DDQN achieves 98.31% of the optimal when $\epsilon$ = 0.5 and 89.1% of the optimal when $\epsilon$ = 0.05.

It can be seen that the DDQN algorithm performs better when the value of the exploration–exploitation trade-off ($\epsilon$) during evaluation is 0.5 (achieving 96.56% of the optimal median SE and 98.31% of the optimal mean SE), compared to when $\epsilon$ is 0.05 (achieving 89.1% of the optimal mean SE and 82.85% of the optimal median SE). Using a higher $\epsilon$ implies that the agent takes more random actions, compensating for under-training and helping it to come out of local minima.

For 8 MBSs per km$^2$ (two MBSs in the simulation area), the CDF of median and mean SE achieved by the algorithms are shown in Figure 7a,b, respectively. Again, the DDQN performance is better when using $\epsilon$ = 0.5 (achieving 93.46% of the optimal median SE and 95.83% of the optimal mean SE), than when using an $\epsilon$ = 0.05 (achieving 79.73% of the optimal median SE and 82.89% of the optimal mean SE). The greedy algorithm achieves

the optimal results in the scenarios that we consider. The time complexity of the exhaustive search is exponential in the number of MBSs and UABSs and is given by:

$$\mathcal{C}_{\text{time}} = \mathcal{O}\left(\left(\frac{1}{\Delta_\alpha + 1}\right)^{N_{\text{mbs}}} \times (1/\Delta_\beta + 1)^{N_{\text{mbs}}} \times \left(\frac{\rho_{\text{high}} - \rho_{\text{low}}}{\Delta_\rho}\right)^{N_{\text{mbs}}}\right.$$
$$\left.\times \left(\frac{\rho'_{\text{high}} - \rho'_{\text{low}}}{\Delta'_\rho}\right)^{N_{\text{uabs}}} \times \left(\frac{l^2}{\Delta_x}\Delta_y\right)^{N_{\text{uabs}}}\right). \tag{18}$$

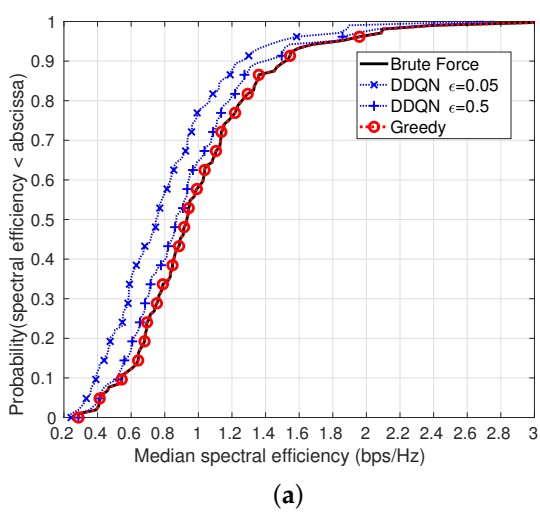

(a)

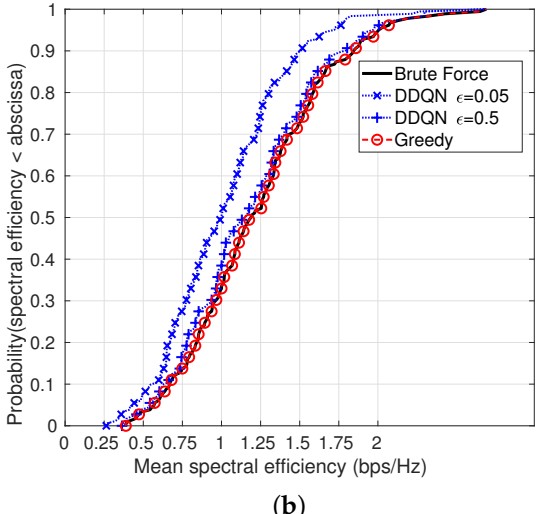

(b)

**Figure 7.** SE results for an MBS density of 8 MBS per km$^2$, which corresponds to two MBSs in the simulation area. (**a**) CDF of median SE. DDQN achieves 93.46% of the optimal when $\epsilon = 0.5$ and 79.73% of the optimal when $\epsilon = 0.05$. (**b**) CDF of mean SE. DDQN achieves 95.83% of the optimal when $\epsilon = 0.5$ and 82.89% of the optimal when $\epsilon = 0.05$.

Unlike the brute force algorithm, the greedy algorithm has linear time complexity, expressed as:

$$\mathcal{O}(N_{\text{mbs}}) + \mathcal{O}(N_{\text{uabs}}). \tag{19}$$

The time complexity of the DDQN approach during training is a function of the number of training steps,

$$\mathcal{O}(N_{\text{train}}),\qquad(20)$$

where $N_{\text{train}}$ is the number of training steps. However, after training is complete and the resultant neural network is used to optimize UAV HetNet parameters, the time complexity is the number of steps in the evaluation phase:

$$\mathcal{O}(N_{\text{eval}}),\qquad(21)$$

where $N_{\text{eval}}$ is the number of evaluation steps. To optimze all parameters for a realization consisting of two MBSs and one UABS, considering the range of parameters given in Table 5, the brute force algorithm searches over $409,600$ parameter combinations, the greedy approach evaluates 464 parameter combinations, while the searches over 500 parameter combinations, as per the policy learned after training.

## 7. Conclusions

We studied how 3GPP LTE FeICIC parameters can be tuned in a UAV-HetNet to maximize the mean and median spectral efficiency of a cellular network. Computationally efficient AI and greedy approaches were compared with an optimal exhaustive search. The AI approach was implemented using DDQN RL with a single agent to model all UAVs and BSs in the network. We observed that, for the scenarios that we considered, the greedy algorithm achieved the optimal mean and median SE, while the AI approach achieved 93.46% of the optimal median SE and 95.83% of the optimal mean SE.

**Author Contributions:** Formal analysis, S.S., A.K., İ.G., M.L.S.; Investigation, S.S., A.K., İ.G., M.L.S.; Methodology, S.S., A.K., İ.G., M.L.S.; Writing—original draft, S.S.; Writing—review & editing, İ.G. and M.L.S. All authors have read and agreed to the published version of the manuscript.

**Funding:** This research was funded in part by NSF under the grant numbers CNS-1453678 and CNS-1738093.

**Conflicts of Interest:** The authors declare no conflict of interest.

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
