# Peer review of "Intelligent Interference Management in UAV-Based HetNets"

_telecom, doi:10.3390/telecom2040027_

Round 1

Reviewer 1 Report

The authors present a very interesting research work about a hot topic. In general, the text is well written and sections are well organized and easy to follow.

I only have some minor comments about the selection of the machine learning technique they've chosen, why that? I mean, RL and DRL are really relevant nowadays because of their good results, but is there any other reason?

Also for the simulation, the authors explain the parameters used, but more detail on the actual environment deployed (tools, etc.) would be appreciated.

Author Response

We would like to thank the reviewer for helping us improve the quality of our work through their feedback. We edited the draft to address their comments, and all such changes are in blue font color. We hope these modifications make our manuscript suitable for publication. Below are our responses to the review comments.

Comment 1: I only have some minor comments about the selection of the machine learning technique they've chosen, why that? I mean, RL and DRL are really relevant nowadays because of their good results, but is there any other reason?

Response: Since RL techniques are relevant and known to be effective for large search spaces, is intructive to investigate their performance and suitability for solving the UAV HetNet optimization problem.

Comment 2: Also for the simulation, the authors explain the parameters used, but more detail on the actual environment deployed (tools, etc.) would be appreciated 
Response: Section. 6 ``Simulation Results" has been updated to include more detail on the simulation tools and libraries used.

Reviewer 2 Report

The authors focus their study on the impact of the unmanned aerial vehicles locations and other cellular parameters on the problem of maximizing the spectral efficiency of a network. Specifically, the authors compare machine learning techniques to optimization approaches in order to find the way the UAV locations and other cellular parameters can be optimized in order to maximize the spectral efficiency of the network.

The manuscript is well-written and easy to follow and the authors have well thought out their main contributions. The provided theoretical analysis is concrete, complete, and correct and the authors have provided all the intermediate steps in order to enable the reader to easily follow it.

The provided numerical results are rich in order to show the pure operation and the performance of the proposed framework. The authors should consider the following suggestions provided by the reviewer in order to improve the scientific depth of their manuscript, as well as they should consider the following comments in order to improve the quality of presentation of their manuscript.

Initially, the provided related work is quite limited for a journal paper publication and the author should elaborate more on the recent advances in the field, such as Mozaffari, M., Saad, W., Bennis, M., & Debbah, M. (2017). Mobile unmanned aerial vehicles (UAVs) for energy-efficient Internet of Things communications. IEEE Transactions on Wireless Communications, 16(11), 7574-7589, Sikeridis, D., et al. "Wireless powered Public Safety IoT: A UAV-assisted adaptive-learning approach towards energy efficiency." Journal of Network and Computer Applications 123 (2018): 69-79, that they either follow traditional machine learning approaches or game theoretic approaches, respectively.

Furthermore, the authors should include an additional subsection in their manuscript discussing the computational complexity of the proposed approaches and they should provide some additional numerical results quantifying the computational complexity with realistic metrics.

Finally, the overall manuscript should be checked for typos, syntax, and grammar errors in order to improve the quality of its presentation.

Author Response

We would like to thank the reviewer for helping us improve the quality of our work through their feedback. We edited the draft to address their comments, and all such changes are in blue font color. We hope these modifications make our manuscript suitable for publication. Below are our responses to the review comments.

Comment 1: The provided related work is quite limited for a journal paper publication and the author should elaborate more on the recent advances in the field, such as Mozaffari, M., Saad, W., Bennis, M., & Debbah, M. (2017). Mobile unmanned aerial vehicles (UAVs) for energy-efficient Internet of Things communications. IEEE Transactions on Wireless Communications, 16(11), 7574-7589, Sikeridis, D., et al. "Wireless powered Public Safety IoT: A UAV-assisted adaptive-learning approach towards energy efficiency." Journal of Network and Computer Applications 123 (2018): 69-79, that they either follow traditional machine learning approaches or game theoretic approaches, respectively.

Response: We updated Section 2 "Literature Review" to include the suggested references, and also included a few other related work that were published within the past three years.

Comment 2: The authors should include an additional subsection in their manuscript discussing the computational complexity of the proposed approaches and they should provide some additional numerical results quantifying the computational complexity with realistic metrics. 
Response: Section. 6 "Simulation Results" has been updated to include a paragraph comparing the complexity of the proposed algorithms. More detailed analysis would consume time and is possible if we are granted an additional two weeks to run simulations and collect data on computational complexity. 

Comment 3: The overall manuscript should be checked for typos, syntax, and grammar errors in order to improve the quality of its presentation.

Response: We apologize for the errors. We went through the manuscript to correct such errors.

Round 2

Reviewer 2 Report

Reviewers' comments have been addressed. The references are not presented in the correct format -- The majority of the journals miss the volume, number, and pages. References 2, 3, 15, 28 are published where? Those points need to be fixed.

Author Response

Thank you for your comments. We re-formated the reference section, and below are our comments:

  1. References to web-pages (References 2, 3, 15, & 28) have been updated to specify that they are online. Their date of accesses and url have also been included. 
  2. The volume number for Reference 13 was given to be the same as its published date (2014): https://jwcn-eurasipjournals.springeropen.com/articles/10.1186/1687-1499-2014-189
  3. Reference 14 is available as an arxiv preprint.
  4. Reference 17 has not been published yet, and so the volume and page number of its journal are not available. Its pre-print is available and it has been assigned a DOI, which we cited: https://www.computer.org/csdl/journal/tm/5555/01/09456090/1usgjVG53fW
  5. We included volume numbers for references 25, 23
  6. Reference 29: The page number for the journal is not available with the publisher